# Heme-Mediated Activation of the Nrf2/HO-1 Axis Attenuates Calcification of Valve Interstitial Cells

**DOI:** 10.3390/biomedicines9040427

**Published:** 2021-04-15

**Authors:** Enikő Balogh, Arpan Chowdhury, Haneen Ababneh, Dávid Máté Csiki, Andrea Tóth, Viktória Jeney

**Affiliations:** 1MTA-DE Lendület Vascular Pathophysiology Research Group, Research Centre for Molecular Medicine, Faculty of Medicine, University of Debrecen, 4032 Debrecen, Hungary; balogh.eniko@med.unideb.hu (E.B.); chowdhury.arpan007@gmail.com (A.C.); hababneh7@gmail.com (H.A.); csiki.david.mate@gmail.com (D.M.C.); andrea.toth@med.unideb.hu (A.T.); 2Doctoral School of Molecular Cell and Immune Biology, Faculty of Medicine, University of Debrecen, 4032 Debrecen, Hungary

**Keywords:** valve calcification, valve interstitial cell (VIC), osteogenic differentiation, heme, Nrf2, heme oxygenase-1 (HO-1), ferritin

## Abstract

Calcific aortic valve stenosis (CAVS) is a heart disease characterized by the progressive fibro-calcific remodeling of the aortic valves, an actively regulated process with the involvement of the reactive oxygen species-mediated differentiation of valvular interstitial cells (VICs) into osteoblast-like cells. Nuclear factor erythroid 2-related factor 2 (Nrf2) regulates the expression of a variety of antioxidant genes, and plays a protective role in valve calcification. Heme oxygenase-1 (HO-1), an Nrf2-target gene, is upregulated in human calcified aortic valves. Therefore, we investigated the effect of Nrf2/HO-1 axis in VIC calcification. We induced osteogenic differentiation of human VICs with elevated phosphate and calcium-containing osteogenic medium (OM) in the presence of heme. Heme inhibited Ca deposition and OM-induced increase in alkaline phosphatase and osteocalcin (OCN) expression. Heme induced Nrf2 and HO-1 expression in VICs. Heme lost its anti-calcification potential when we blocked transcriptional activity Nrf2 or enzyme activity of HO-1. The heme catabolism products bilirubin, carbon monoxide, and iron, and also ferritin inhibited OM-induced Ca deposition and OCN expression in VICs. This study suggests that heme-mediated activation of the Nrf2/HO-1 pathway inhibits the calcification of VICs. The anti-calcification effect of heme is attributed to the end products of HO-1-catalyzed heme degradation and ferritin.

## 1. Introduction

Calcific aortic valve stenosis (CAVS) is a heart disease characterized by a progressive fibro-calcific remodeling and thickening of the aortic valves eventually leading to severe heart outflow tract obstruction [1]. CAVS is the second-most frequent cardiovascular disease, with a prevalence of 0.4% in the general population. It is also considered an aging disease, as the prevalence of severe CAVS increases to 1.7% in the population over 65 years old [1]. 

Until recently, CAVS was thought to be a passive, degenerative process, but this assumption was challenged by the detection of osteoblast-like and osteoclast-like cells in human aortic valve leaflets [2]. Now, CAVS is considered to be an actively regulated process in which differentiation of valvular interstitial cells (VICs) into osteoblast-like cells and myofibroblasts occurs [3,4]. 

Under in vitro conditions, VICs respond to osteogenic stimuli through the upregulation of several osteochondrogenic markers, including runt-related transcription factor 2 (RUNX2), bone morphogenetic protein 2 (BMP2), Sry-related HMG box-9 (SOX9), alkaline phosphatase (ALP), osteopontin (OPN), and osteocalcin (OCN), and these markers have been detected in human diseased aortic valves [5,6]. 

Patients with chronic kidney disease (CKD) are characterized by accelerated and premature cardiovascular calcification, which is a clear exhibition of the CKD-associated premature aging phenotype [7,8]. In CKD patients, the prevalence of severe CAVS ranges between 6-13%, a rate significantly higher than in the general population [9,10]. As a consequence, CKD patients exhibit extremely high cardiovascular mortality, similar to that seen in the elderly (≥75) general population [11,12]. 

CKD is associated with the dysregulation of calcium (Ca) and phosphate (P) metabolism, leading to hyperphosphatemia and high-circulating Ca x P products [13]. Increased P and Ca synergistically stimulate osteochondrogenic differentiation and the extracellular matrix (ECM) calcification of vascular smooth muscle cells (VSMCs) [13,14,15]. In CKD patients, higher serum P concentrations are associated with an increased prevalence of vascular and valvular calcification, suggesting a causative role of high P in CKD-associated calcification [16]. 

The elevation of reactive oxygen species (ROS) formation plays a role in both vascular and valve calcification [17,18]. Normally, ROS formation is counterbalanced by a complex antioxidant defense system. Nuclear factor erythroid 2-related factor 2 (Nrf2) is an important element of this antioxidant network [19]. Under homeostasis, Nrf2 binds to its negative regulator, Kelch-like ECH-associated protein 1 (Keap1) in the cytosol [19]. This interaction initiates the polyubiquitinylation and proteasomal degradation of Nrf2. Different oxidants and electrophiles induce the modification of cysteine residues of Keap1, leading to the disruption of the Nrf2–Keap1 interaction, Nrf2 stabilization, and nuclear translocation. Once in the nucleus, Nrf2 binds to the antioxidant response elements located at the promoter regions of a variety of antioxidant genes [19]. Previous studies have shown that upregulation of the Nrf2 system attenuates the high P-induced calcification of VSMCs [20,21,22]. 

Under homeostasis, heme is an essential molecule that serves as a prosthetic group of diverse hemoproteins [23]. In contrast, labile heme is potent pro-oxidant and pro-inflammatory molecule [24,25,26]. Labile heme promotes Nrf2 stabilization via a Keap1-dependent signaling mechanism and regulates the transcription of heme oxygenase-1 (HO-1), the enzyme responsible for heme catabolism [27,28]. Heme catabolism by HO-1 is protective against the deleterious effects of labile heme and provides protection from hemolytic conditions and polymicrobial sepsis [29,30,31]. The mechanism through which heme catabolism by HO-1 confers protection in diverse diseases is not completely understood, but it has been linked to the generation of the heme degradation products carbon monoxide (CO), biliverdin, and iron [23]. Because HO-1 has been found to be upregulated in human calcified aortic valves [32], we investigated the effect of the Nrf2/HO-1 pathway activation on P- and Ca-induced osteogenic differentiation and ECM calcification of VICs. 

## 2. Materials and Methods

### 2.1. Cell Culture

Human VICs were obtained from Innoprot (Derio, Spain). Cells were maintained in Dulbecco’s modified Eagle’s medium (DMEM, D6171, Sigma, St. Louis, MO, USA) supplemented with 10% FBS (10270-106, Gibco, Grand Island, NY, USA), antibiotic antimycotic solution (A5955, Sigma, St. Louis, MO, USA), sodium pyruvate (S8636, Sigma, St. Louis, MO, USA), and L-glutamine (G7513, Sigma, St. Louis, MO, USA). Cells were maintained at 37 °C in a humidified atmosphere containing 5% CO_2_. Cells were grown to confluence and used from passages 5 to 8. 

### 2.2. Induction of Osteogenesis

At confluence, VICs were switched to the osteogenic medium, which was prepared by adding inorganic phosphate (P) as a mixture of NaH_2_PO_4_ and Na_2_HPO_4_, pH 7.4 (0–2.5 mmol/L), and Ca in the form of CaCl_2_ (0.3–1.2 mmol/L) to the growth medium. Both the growth and osteogenic media were changed every three days. Unless otherwise specified, we used an osteogenic medium that was supplemented with 2.5 mmol/L P and 0.3 mmol/L Ca. 

### 2.3. Cell Treatments

Iron was introduced as ammonium ferric citrate (F5879, Sigma, St. Louis, MO, USA), dissolved in deionized water. Heme (H9039, Sigma, St. Louis, MO, USA) was dissolved in NaOH (20 mmol/L). Tin protoporphyrin IX (SnPP, 16375, Cayman Chemical, Ann Arbor, MI, USA) and zinc protoporphyrin IX (ZnPP, 691550-M, EMD Millipore Corp., Burlington, MA, USA) were dissolved in DMSO. The final concentration of NaOH was kept below 2 mmol/L, and the DMSO was less than 1% in all experiments. To deliver CO, we used the tricarbonyl-dichloro-ruthenium (II) dimer also known as CO-releasing molecule 2 (CORM2), [Ru_2_Cl_4_(CO)_6_] (288144, Sigma, St. Louis, MO, USA). CORM2 was dissolved in DMSO immediately before use, and it was administered every 12 h. The Nrf2 inhibitor ML385 (SML1833, Sigma, St. Louis, MO, USA) was dissolved in DMSO. Ferritin (FT) was administered as holoferritin (F4503, Sigma, St. Louis, MO, USA). 

### 2.4. Alizarin Red (AR) Staining and Quantification

After washing with Dulbecco’s PBS (DPBS; D8537, Sigma, St. Louis, MO, USA), the cells were fixed in 4% paraformaldehyde (16005, Sigma, St. Louis, MO, USA) and rinsed with deionized water thoroughly. Cells were stained with Alizarin Red S (A5533, Sigma, St. Louis, MO, USA) solution (2%, pH 4.2) for 20 min at room temperature. Excessive dye was removed through several washes in deionized water. To quantify AR staining in 96-well plates, we added 100 µL of hexadecyl-pyridinium chloride (C9002, Sigma, St. Louis, MO, USA) solution (100 mmol/L) to the wells and measured the optical density (OD) with a microplate reader (800 TS, Biotek, Winooski, VT, USA) at 560 nm using hexadecyl-pyridinium chloride solution as a blank. 

### 2.5. Quantification of Ca Deposition

Cells grown on 96-well plates were washed twice with DPBS, and decalcified with HCl (30721, Sigma, St. Louis, MO, USA, 0.6 mol/L) for 30 min at room temperature. The Ca content of the HCl supernatants was determined using the QuantiChrome Calcium Assay Kit (DICA-500, Gentaur, Kampenhout, Belgium). Following decalcification, cells were washed twice with DPBS, and solubilized with a solution of NaOH (S8045, Sigma, St. Louis, MO, USA, 0.1 mol/L) and sodium dodecyl sulfate (11667289001, Sigma, St. Louis, MO, USA 0.1%), and the protein content of the samples was measured with the BCA protein assay kit (23225, Pierce Biotechnology, Rockford, IL, USA). The Ca content of the cells were normalized to the protein content, and expressed as mg/mg protein.

### 2.6. Quantification of OCN

For OCN detection, the ECM of the cells grown on 6-well plates was dissolved in 100 µL of EDTA (E6758, Sigma, St. Louis, MO, USA, 0.5 mol/L, pH 6.9). The OCN content of the EDTA-solubilized ECM samples was quantified by an enzyme-linked immunosorbent assay (DY1419-05, DuoSet ELISA, R&D, Minneapolis, MN, USA), used according to the manufacturer’s protocol.

### 2.7. Determination of Cell Viability

Cell viability was determined by the MTT assay, as previously described [19]. Briefly, following the treatments, the cells in the 96-well plates were washed with PBS and 100 µL of 3-[4,5-Dimethylthiazol-2-yl]-2,5-diphenyl-tetrazolium bromide (MTT, ML2128, Sigma, St. Louis, MO, USA, 0.5 mg/mL) solution was added. After a 4 h of incubation in the cell culture incubator, the MTT solution was removed, the formazan crystals were dissolved in 100 µL of DMSO, and the optical density was measured at 570 nm.

### 2.8. Quantitative RT-PCR

RNA was isolated from cells using TRIzol (CS502, RNA-STAT60, Tel-Test Inc., Friendswood, TX, USA), according to the manufacturer’s protocol. Two micrograms of RNA were reverse-transcribed to cDNA using High-Capacity cDNA Reverse Transcription Kit (4368813, Applied Biosystems, Waltman, MA, USA). For the measurement of the mRNA levels, the reaction mixture contained 0.1 µg reverse-transcribed sample, 10 µmol/L of forward (5’-GAGACAGGTGAATTTCTCCCAAT-3’) and reverse (5’GGGAGTAGTTGGCAGATCCA-3’) primers for Nrf2, forward (5’-TTCAGAAGGGCCAGGTGA-3’) and reverse (5’-TGTTGCGCTCAATCTCCTC-3’) primers for HO-1, forward (5’- TACCCGCACTTGCACAAC-3’) and reverse (5’- TCTCGCTCTCGTTCAGAAGTC-3’) primers for SOX9, forward (5’-AGCTGGATGACCAGAGTGCT-3’) and reverse (5’- GCTCTCATCATTGGCTTTCC-3’) primers for osteopontin (OPN) and 5 µL of LightCycler 480 SYBR GREEN I Master Mix (04887352001, Bio-Rad Laboratories, Hercules, CA, USA). PCRs were carried out using the Real-Time PCR System (Bio-Rad Laboratories, Hercules, CA, USA). Relative mRNA expressions were calculated with the ΔΔCt method, using HPRT as internal control.

### 2.9. Western Blot

For the evaluation of the ALP, RUNX2, HO-1, Nrf2, ferritin H-chain (FtH), and ferritin L-chain (FtL) protein expressions, cell lysates were electrophoresed in 10% SDS-PAGE, then blotted onto a nitrocellulose membrane (1060003, Amersham Proton, GE Healthcare, Chicago, IL, USA). Western blotting was performed with the use of anti-ALP antibody (sc30203, Santa Cruz Biotechnology, Inc, Dallas, TX), anti-RUNX2 antibody (GTX81326, GeneTex, Irvine, CA, USA) at a 1:1000 dilution, anti-HO-1 antibody (70081, Cell Signaling Technology, Leiden, The Netherlands) at a 1:2000 dilution, anti-Nrf2 antibody (16396-1-AP, Proteintech, Rosemont, IL, USA) at a 1:1000 dilution, anti-FTH antibody (4393, Cell Signaling Technology, Leiden, Netherlands) at a 1:1000 dilution, anti-FTL antibody (ab69090, Abcam, Cambridge, United Kingdom) at a 1:500 dilution, followed by the HRP-labeled anti-rabbit or anti-mouse IgG secondary antibodies (NA-934 and NA-931, Amersham Biosciences Corp., Piscataway, NJ, USA). Antigen-antibody complexes were detected by enhanced chemiluminescence using ClarityTM Western ECL Substrate (170-5060, Bio-Rad Laboratories, Hercules, CA, USA). Chemiluminescent signals were detected conventionally on an X-ray film or digitally by using a C-Digit Blot Scanner (LI-COR Biosciences, Lincoln, NE, USA). After detection, the membranes were stripped and reprobed for β-actin using anti-β-actin antibody at a dilution of 1:2000 (sc-47778, Santa Cruz Biotechnology Inc., Dallas, TX, USA). Blots were quantified by using the inbuilt software on the C-Digit Blot Scanner (LI-COR Biosciences, Lincoln, NE, USA).

### 2.10. Statistics

Results are expressed as mean ± SD. At least three independent experiments were performed for all in vitro studies. Statistical analyses were performed with GraphPad Prism software (v.8.01, San Diego, CA, USA). The Shapiro–Wilk test was performed to assess the normality of the distribution. All data passed the normality and equal variance tests, therefore parametric tests were used to determine *p* values. Statistically significant differences between two groups were assessed using a two-tailed Student’s *t*-test. Comparisons between more than two groups were carried out by one-way ANOVA followed by Tukey’s multiple comparisons test. To compare each of a number of treatment groups with a single control group, we performed a one-way ANOVA followed by Dunnett’s post hoc test. A value of *p* < 0.05 was considered significant.

## 3. Results

### 3.1. Phosphate and Ca Synergistically Induce ECM Mineralization and Cell Death in VICs

To set up an in vitro model of aortic valve calcification, we used human primary VICs that morphologically displayed characteristics of both fibroblasts and smooth muscle cells (Figure 1a). To induce calcification, we cultured human VICs in a calcification medium that was supplemented with different concentrations of P (0–2.5 mmol/L) and Ca (0–1.2 mmol/L). ECM calcification was evaluated by AR staining after 7 days of treatment (Figure 1b,c). We found that P and Ca synergistically and dose-dependently induced ECM calcification in VICs. Then we measured Ca levels in HCl-solubilized ECM samples, which confirmed the results of the AR staining (Figure 1d). The combination of 2.5 mmol/L P and 0.3 mmol/L Ca resulted in about a 5-fold elevation in the ECM Ca content over the control (0.20 ± 0.03 vs. 1.10 ± 0.13 mg Ca/mg protein) (Figure 1d), whereas the 2.0 mmol/L P and 0.6 mmol/L media triggered an about 8.5-fold increase. The calcification of VIC cells reached its maximum in the presence of the 0.6 mmol/L Ca and 2.5 mmol/L P media (Figure 1d). A further increase in the amount of Ca resulted the loss of P-sensitivity and yielded maximum calcification regardless of the P concentration (Figure 1d). Previously, calcification of VICs has been associated with apoptotic cell death [33], therefore next we investigated the effect of P- and Ca-driven osteogenic stimulation on cell viability. We observed that excess P and Ca triggered cell death synergistically in a dose-dependent manner, resulting in a substantial decrease in the viability of VICs when the Ca concentration exceeded 0.6 mmol/L and the P levels were higher than 2 mmol/L (Figure 1e). These results suggest that excess P and Ca induce cell death and Ca deposition in the ECM of VICs. RUNX2 is the master transcription factor of osteogenic differentiation. Therefore, we investigated whether osteogenic stimuli (OM) upregulated RUNX2 expression. We found that OM (2.5 mmol/L Pi and 0.3 mmol/L Ca) increased RUNX2 expression about 4-fold (Figure 1f).

### 3.2. Heme Inhibits P- and Ca-Induced ECM Calcification and Osteogenic Transdifferentiation of VICs 

To induce VIC calcification without triggering massive cell death, we set up the P and Ca concentrations to 2.5 mmol/L and 0.3 mmol/L respectively, and used this supplementation to the osteogenic medium (OM) in all further experiments. To investigate the effect of heme on VIC calcification, we treated the VICs with OM in the presence of various concentrations of heme (1–50 µmol/L). First, we assessed ECM calcification by AR staining and observed that heme strongly inhibited VIC calcification in a dose-dependent manner (Figure 2a). Heme at a concentration of 10 µmol/L reached its maximal inhibitory effect based on AR staining quantification (Figure 2a). Then we evaluated how heme influenced ECM Ca levels. Heme at a concentration of 1 µmol/L already inhibited OM-induced calcification significantly (2.02 ± 0.10 vs. 1.21 ± 0.31 mg Ca/mg protein) and higher concentrations have even more profound effects (Figure 2b). 

Osteogenic stimulation triggers the trans-differentiation of VICs to osteoblast-like cells. This process can be monitored by the detection of certain osteoblast-specific proteins including ALP and OCN. Therefore, next we investigated the effect of heme on the expression of these markers. Osteogenic stimulation with OM triggered an about 1.5-fold increase in the expression of ALP as compared to control VICs (GM) (Figure 2c). This increase was blunted in the presence of heme at the concentration of 5, 10, and 25 µmol/L (Figure 2c). Next, we determined the level of OCN, a typical Ca-binding non-collagenous bone-matrix protein, in the ECM of the VICs. The OCN level of OM-treated VICs were about 10-fold higher than in the controls (GM) (Figure 2d). This increase was completely diminished in the presence of heme at a concentration of 10 and 25 µmol/L (Figure 2d).

### 3.3. Induction of the Nrf2/HO-1 Axis by Heme in VICs

Nrf2, the transcription factor that controls the expression of numerous antioxidant genes, has been implicated in vascular pathologies. Based on previous studies, upregulation of the Nrf2 system attenuates high P-induced calcification of VSMCs [20,21,22]. Because heme is a known inducer of Nrf2 in various cell types, first we investigated whether heme induces the expression of Nrf2 in VICs. Heme at a concentration of 50 µmol/L induced an about 4-fold increase in the Nrf2 mRNA level (4 h) (Figure 3a). Nrf2 regulates the expression of HO-1, a protein with diverse anti-oxidant and anti-inflammatory actions. Therefore, we investigated whether the heme-mediated upregulation of Nrf2 was accompanied by increased expression of HO-1. As we expected, heme induced a strong and dose-dependent upregulation of HO-1 mRNA (4 h) expression in VICs (Figure 3b). Then we assessed the Nrf2 and HO-1 protein expressions (12 h) in heme-treated VICs. Every applied concentration of heme (5–50 µmol/L) triggered an about 2.5-fold increase in the expression of Nrf2 compared to the control (GM) (Figure 3c). Additionally, we observed a strong and dose-dependent upregulation of HO-1 in heme-treated VICs (Figure 3c). 

### 3.4. Anti-Calcification Effect of Heme Requires Integrity of the Nrf2/HO-1 Axis

We investigated whether the induction of the Nrf2/HO-1 system played a critical role in the heme-mediated inhibition of VIC calcification. Our first approach was to inhibit the transcriptional activity of Nrf2 with the well-characterized inhibitor ML385. We induced calcification of the VICs with OM in the presence or absence of heme (10 µmol/L) and in the presence or absence of ML385 (10 µmol/L). To assess calcification, we performed AR staining on day 5 (Figure 4a). As we expected, heme completely inhibited OM-induced VIC calcification in the absence of ML385 (Figure 4a). In contrast, in the presence of ML385, heme lost its ability to inhibit the OM-induced calcification of VICs (Figure 4a), suggesting that the transcriptional activity of Nrf2 was necessary for the anti-calcification effect of heme. Moreover, we found that OM triggered stronger calcification in the presence of ML385, highlighting the protective role of Nrf2 in P and Ca-induced VIC calcification (Figure 4a). We confirmed these results with the evaluation of ECM Ca levels (Figure 4b). We found higher Ca levels in the ECM of cells that were treated with ML385, heme, and OM compared to the cells that were treated with ML385 and OM (2.66 ± 0.54 vs. 1.66 ± 0.12 mg Ca/mg protein) (Figure 4b). 

Then we investigated the role of HO-1 in the anti-calcification effect of heme. We induced calcification of the VICs with OM in the presence of heme and the pharmacological inhibitors of HO-1 enzyme activity SnPP and ZnPP. Heme (10 µmol/L) completely inhibited OM-induced ECM calcification in the absence of HO-1 inhibitors (Figure 4c). In contrast, in the presence of either SnPP (10 µmol/L) or ZnPP (10 µmol/L), heme lost its ability to inhibit OM-induced VIC calcification (Figure 4c). To confirm these results, we measured the Ca content of HCl-solubilized ECM. The Ca level of heme-treated VIC was decreased significantly compared to those of the OM-treated cells, and this decrease was completely lost in the presence of HO-1 inhibitors (Figure 4d). These results suggests that intact HO-1 enzyme activity is needed for the anti-calcification effect of heme.

### 3.5. Heme Degradation Products Possess Anti-Calcification Activities

The inhibition of HO-1 activity impairs the anti-calcification effect of heme, which suggests the possibility that heme degradation products are responsible for the inhibitory effects of heme. HO-1-catalized heme degradation yields biliverdin, which is promptly converted to bilirubin (BR) by biliverdin reductase (BVR), iron (Fe), and CO (Figure 5a). Therefore, next we investigated the effect of BR, Fe, and CO on VIC calcification. We induced ECM calcification in the VICs with OM in the presence or absence of BR (5–50 µmol/L), Fe (5–50 µmol/L), and CO administered as CORM2 (5–50 µmol/L). As shown by AR staining performed on day 5, all the three heme degradation products exhibited dose-dependent anti-calcification potential (Figure 5b–d). We confirmed these results by measuring the Ca content of solubilized ECM. BR, Fe, and CO inhibited osteogenic stimulation-induced Ca accumulation in the ECM of VICs. BR at a 50 µmol/L concentration completely inhibited OM-induced calcification, whereas Fe and CO provided partial protection (Figure 5e–g). 

Next, we investigated the effect of the heme degradation products on the levels of OCN in the ECMs of OM-stimulated VICs (Figure 5h). Osteogenic stimulation (OM) triggered an about 80-fold increase of OCN in the ECM of VICs as compared to controls (GM) (0.037 ± 0.016 vs. 3.025 ± 0.304 ng/mg protein) (Figure 5h). This increase was totally abolished in the presence of heme and BR, and it was partially inhibited by Fe and CO at a concentration of 50 µmol/L (Figure 5h). Then, we evaluated the mRNA levels of SOX9, a transcription factor involved in the osteochondrogenic trans-differentiation of VICs, and OPN, a bone-related glycoprotein (Figure 5i,j). Heme and BR inhibited OM-induced increases in both the SOX9 and OPN mRNA expressions, whereas Fe and CO did not exhibit such a protective effect (Figure 5i,j).

### 3.6. Ferritin Mimics the Inhibitory Effect of Heme on VIC Calcification

Heme degradation by HO-1 leads to the release of free iron from the heme moiety and the subsequent upregulation of ferritin (FT), the key molecule in iron storage. Because FT has many beneficial effects in vascular pathology, we investigated its role in VIC calcification. First, we evaluated the protein expressions of both ferritin subunits, FTH (ferritin heavy/heart) and FTL (ferritin light/liver). Heme at a concentration of 5 µmol/L upregulated the expressions of both FTH and FTL by about 15-fold and 20-fold, respectively (Figure 6a). Then we addressed whether ferritin could inhibit OM-induced VIC calcification. We administered ferritin as a holoprotein and assessed calcification by AR staining and ECM Ca measurement. Both methods demonstrated that holo-FT completely inhibits OM-induced calcification at a concentration of 100 µg/mL (Figure 6b,c).

## 4. Discussion

Nrf2 is encoded by the nuclear factor erythroid-derived 2-like 2 gene, a basic-leucine zipper-like transcription factor belonging to the Cap‘n’Collar subfamily [34]. Nrf2 is a stress-responsive transcription factor that regulates the expression of more than 250 genes encoding cytoprotective, antioxidant, and phase II-detoxifying enzymes in order to limit oxidative stress and maintain cellular homeostasis [34]. 

HO-1 is one of the classic Nrf2-regulated genes. HO-1 catalyzes the degradation of heme, a molecule with pro-oxidant and pro-inflammatory properties, into compounds with anti-oxidant and anti-inflammatory features, such as CO and bilirubin [23]. Heme degradation by HO-1 is associated with the upregulation of ferritin, the key iron storage protein that keeps liberated heme-iron in a redox inactive state [35]. Recently, Olkowitz et al. reported increased expression of HO-1 in calcified aortic valves [32], therefore here we investigated the role of the Nrf2/HO-1 pathway in valve calcification. 

The Ca x P product is the most predictive parameter in the prevalence of valve calcification in CKD patients [36]. We therefore used a cellular model of valve calcification in which we induced calcification in VICs with an osteogenic medium containing elevated levels of P and Ca (Figure 1). In agreement with previous reports, we found that elevated level of P and Ca induced ECM calcification in VICs [37]. The effect of P and Ca was dose-dependent and synergistic, and extensive Ca accumulation in the ECM of VICs was associated with cell death (Figure 1). 

To induce the Nrf2/HO-1 pathway, we used heme, the ubiquitous iron compound with well-known deleterious effects in vascular pathology. Heme largely amplifies oxidant-mediated cell death in endothelial cells (ECs), due to its ability to catalyze the Haber–Weiss reaction, in which highly reactive hydroxyl radicals are formed [26,38]. Heme can also harm ECs indirectly by triggering the oxidative modification of low-density lipoprotein [24]. Additionally, heme activates toll-like receptor 4, and the nuclear factor kappa B (NF-κB) signaling pathway in ECs, leading to endothelial activation and dysfunction, characterized by the production of vascular adhesion molecules, pro-inflammatory cytokines, and ROS [39,40,41]. Heme also triggers the activation of the nucleotide-binding domain, the leucine-rich repeat-containing family, and the pyrin domain containing 3 (NLRP3) inflammasome [42]. 

Recent evidence has proven that NF-κB activation and the excessive production of ROS are important mediators of vascular calcification [17,43,44,45,46,47]. Based on these facts, one can assume that heme would trigger vascular calcification. Surprisingly, we found the opposite: heme is a potent inhibitor of VIC calcification (Figure 2). Our results revealed that heme inhibits ECM calcification and the expression of osteoblast markers, ALP, and OCN in OM-stimulated VICs, in a dose-dependent manner (Figure 2). Previously, Zarjou et al. found a similar effect of heme on the P-induced calcification of VSMCs in vitro [48]. Further studies are needed to address the effect of heme on vascular and valvular calcification in vivo. 

The Nrf2/HO-1 pathway is a potent anti-oxidant and anti-inflammatory pathway, and we hypothesized that its induction might explain the anti-calcification effect of heme. To address this possibility, first we demonstrated that heme induces the Nrf2/HO-1 system in VICs (Figure 3). Moreover, we found that heme lost its anti-calcification potential upon the pharmacological inhibition of Nrf2 in VICs (Figure 4). Our data is in agreement with previous reports in which the importance of Nrf2 to VSMCs calcification was demonstrated. There are numerous endogenous, natural, and pharmacological activators of Nrf2. Studies have shown that many different Nrf2 inducers, including hydrogen sulfide, rosmarinic acid, dimethyl fumarate, and resveratrol inhibit high P-induced VSMC calcification [20,49,50,51,52]. The activation of the Nrf2 signaling pathway prevents high P-induced calcification by inducing autophagy and limiting ROS production [22,53]. Additionally, the mitochondria-targeted anti-oxidant mitoquinone attenuates vascular calcification by suppressing oxidative stress and reducing the apoptosis of VSMCs through Nrf2 activation [54]. 

Here we demonstrated that Nrf2-regulated HO-1, the inducible enzyme responsible for heme degradation, is highly upregulated in heme-treated VICs. Because of the protective nature of heme degradation products, we studied whether HO-1 enzyme activity was necessary for the inhibitory action of heme. We observed that heme lost its anti-calcification effects in the presence of the HO-1 inhibitors ZnPP and SnPP (Figure 4). A similar effect was observed on osteoblast mineralization, where heme inhibited the maturation and mineralization of osteoblasts, while a blockade of HO-1 activity by ZnPP antagonized the inhibitory action of heme [55]. Heme degradation by HO-1 yields CO, free iron, and biliverdin, the latter of which is converted promptly to bilirubin by biliverdin reductase. CO is an important signaling molecule, mainly due to its propensity of binding to Fe2^+^ centers in heme proteins [56,57]. The iron released during heme catabolism is a pro-oxidant, but its harmful effect is controlled by the FT, and mainly the FTH, that is co-expressed with HO-1 [58]. A slight elevation of bilirubin, the third product of heme catabolism, is associated with cardiac and vascular protection, and it decreased all-cause mortality in adults [59,60,61]. 

Here we demonstrated that all the three products of heme catabolism attenuated calcium accumulation in the ECM of OM-induced VICs. Among them, bilirubin was the most powerful inhibitor of VIC calcification (Figure 5). Additionally, bilirubin inhibited the OM-induced upregulation of SOX9 and OPN to a similar extent as heme (Figure 5). Previously, Zarjou et al. showed that heme inhibits VSMC calcification and identified iron and FTH as the main regulators of this process [48]. Therefore, we also examined the role of FT in VIC calcification, and found that FT was a powerful anti-calcification agent in VICs, which mimics the inhibitory effects of heme and iron (Figure 6). The concentration of extracellular FT in serum is below 500 ng/mL in healthy adults that far below the FT concentration that we used in this study (1–1000 µg/mL). Extracellular FT is taken up through specific receptors found on liver cells, human lymphocytes, erythroblasts, adipocytes, and on various cell lines [62], but we lack information about FT uptake by VICs. Further studies are needed to investigate whether VICs take up extracellular FT, and to determine whether the inhibitory effect of FT is driven by the iron dissociated from it. 

The phenotype switch of VSMCs and VICs into osteoblast-like cells is the underlying cellular mechanism in vascular and valve calcification, respectively. This process shows remarkable similarities to the differentiation of mesenchymal stem cells (MSCs) into osteoblasts. Growing evidence suggests that the activation of the Nrf2/HO-1 axis influences osteogenesis. Interestingly, Yoon et al. found that proper Nrf2 signaling is required for maintaining the stemness and self-renewal capability of MSCs, and that blocking the nuclear translocation of Nrf2 decreases the osteogenic differentiation potential of MSCs [63]. The overexpression of HO-1 was found to be associated with the increased osteogenic and decreased adipogenic differentiation potential of MSCs [64]. Regarding the effect of iron, Balogh et al. showed that excess iron inhibits the osteogenic differentiation of MSCs through the induction of FT [65], and Zarjou et al. showed that high iron decreases osteoblast activity and ECM calcification by increasing FTH expression and ferroxidase activity [66]. 

## 5. Conclusions and Future Perspective

To conclude, the data reported in the present study suggest that the heme-mediated activation of the Nrf2/HO-1 pathway inhibits the calcification of VICs. Heme triggers the upregulation of HO-1 and, subsequently, FT in VICs. The anti-calcification effect of heme is attributed to the end-products of HO-1-catalyzed heme degradation, and the FT that is induced by the iron released during heme catabolism. 

There are pharmacological options for the upregulation of HO-1. For example a previous study demonstrated that the intravenous administration of heme arginate, a drug used in the treatment of acute porphyrias, induces HO-1 in the human heart [67]. Further studies are needed to investigate whether the pharmacological upregulation of the Nrf2/HO-1 system or the therapeutic application of heme degradation products could inhibit cardiovascular calcification in vivo. 

## Figures and Tables

**Figure 1 biomedicines-09-00427-f001:**
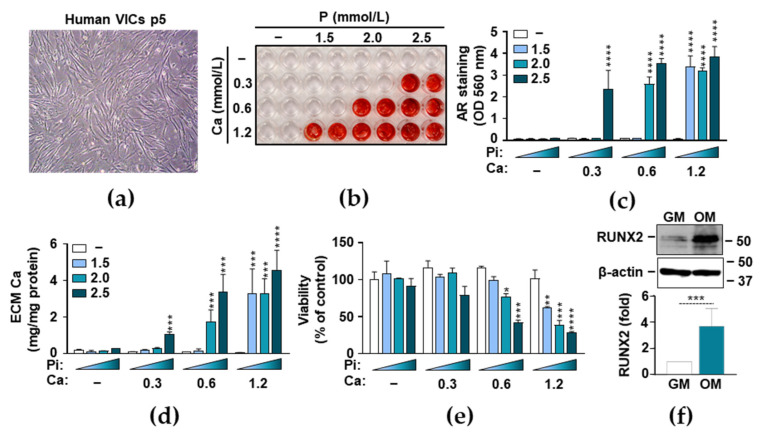
Osteogenic stimuli induce extracellular matrix mineralization and Ca deposition of valve interstitial cells (VICs). (**a**–**f**) Confluent VICs (passage number 5–8) were maintained in growth medium (GM) or in osteogenic medium (OM) obtained by supplementing growth medium (GM) with phosphate (P) (0–2.5 mmol/L) and Ca (0–1.2 mmol/L). (**a**) Bright-field microscope image of human VICs in passage number 5, 100× magnification. (**b**,**c**) Ca deposition in the extracellular matrix (ECM) was evaluated by Alizarin Red (AR) staining after 7 days of treatment. Representative images of stained plates from three independent experiments and quantification are shown. (**d**) Ca content of the HCl-solubilized ECM is presented. (**e**) Cell viability was determined by MTT assay after 7 days of treatment. (**f**) Protein expression of RUNX2 was determined from whole cell lysates (24 h). Membranes were reprobed for β-actin. Representative Western blots and relative expression of RUNX2 normalized to β-actin from three independent experiments are shown. Results are presented as mean ± SD of three independent experiments performed in triplicate. * *p* < 0.05, ** *p* < 0.01, *** *p* < 0.005, **** *p* < 0.001.

**Figure 2 biomedicines-09-00427-f002:**
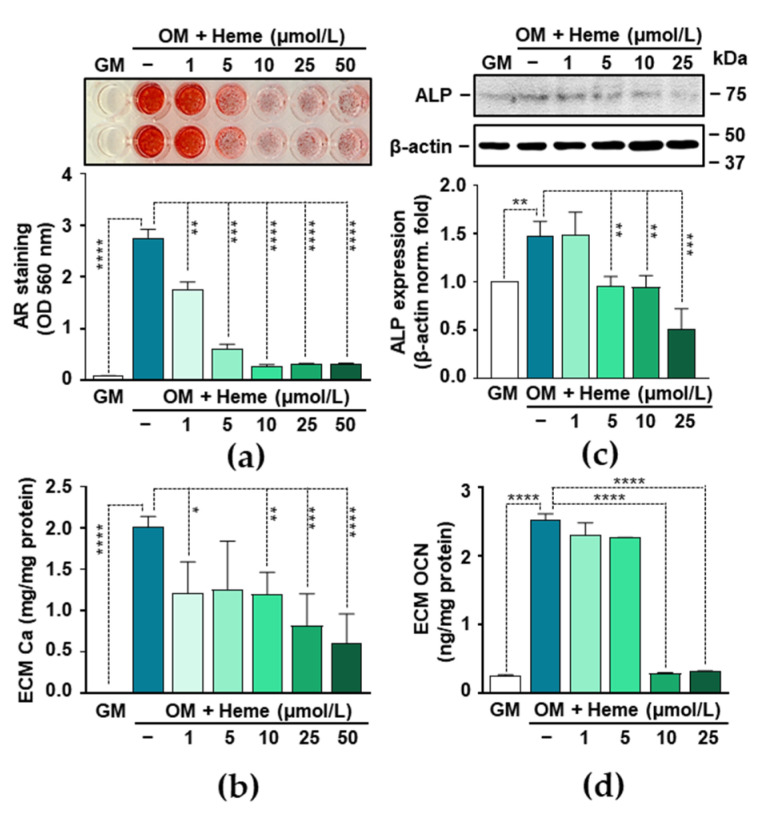
Heme inhibits OM-induced calcification of VICs in a dose-dependent manner. (**a**–**d**) Confluent VICs (passage number 5–8) were maintained in GM or OM (2.5 mmol/L P and 0.3 mmol/L Ca) in the absence or presence of heme (1–50 μmol/L). (**a**) Representative AR staining and quantification (day 5) are shown from three independent experiments. (**b**) Ca content of the HCl-solubilized ECM is presented (day 5). Data are expressed as mean ± SD of three independent experiments performed in triplicate. (**c**) Protein expression of ALP was determined from whole cell lysates (72 h). Membranes were reprobed for β-actin. Representative Western blots and relative expression of ALP normalized to β-actin from three independent experiments are shown. Data are expressed as mean ± SD. (**d**) OCN levels were determined in EDTA-solubilized ECM samples by ELISA (day 5). Results are presented as mean ± SD of three independent experiments performed in duplicate. * *p* < 0.05, ** *p* < 0.01, *** *p* < 0.005, **** *p* < 0.001.

**Figure 3 biomedicines-09-00427-f003:**
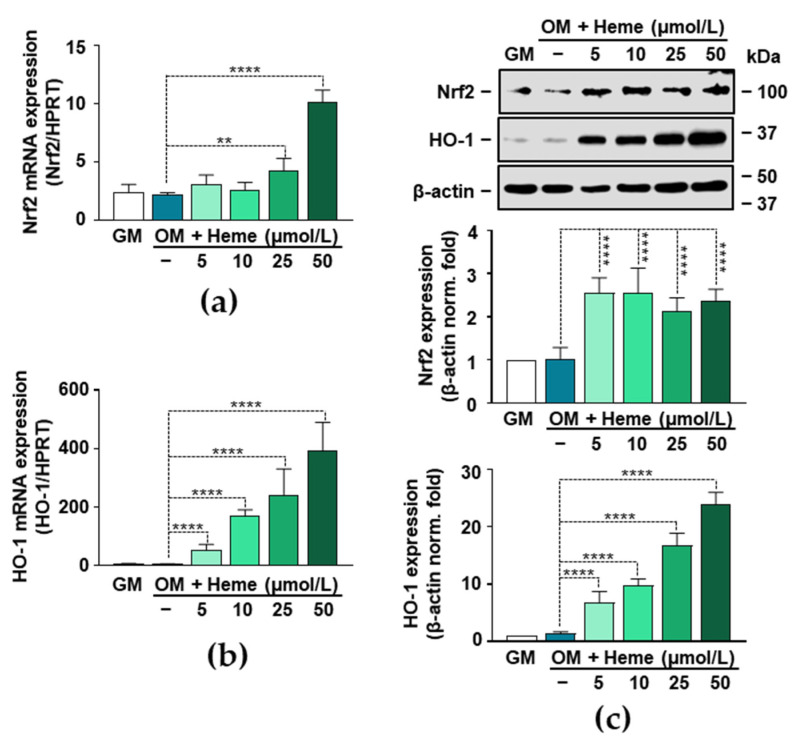
Heme induces Nrf2 and HO-1 in VICs. **(a-c)** VICs (passage number 5–8) were cultured in GM or OM in the absence or presence of heme (5–50 μmol/L). (**a**,**b**) Nrf2 and HO-1 mRNA levels were measured by real time RT-PCR (4 h). (**c**) Protein expression of Nrf2 and HO-1 was determined from whole cell lysates (12 h). Membranes were reprobed for β-actin. Representative Western blots and relative expression of Nrf2 and HO-1 normalized to β-actin from three independent experiments are shown. Results are presented as mean ± SD of three independent experiments performed in triplicate. ** *p* < 0.01, **** *p* < 0.001.

**Figure 4 biomedicines-09-00427-f004:**
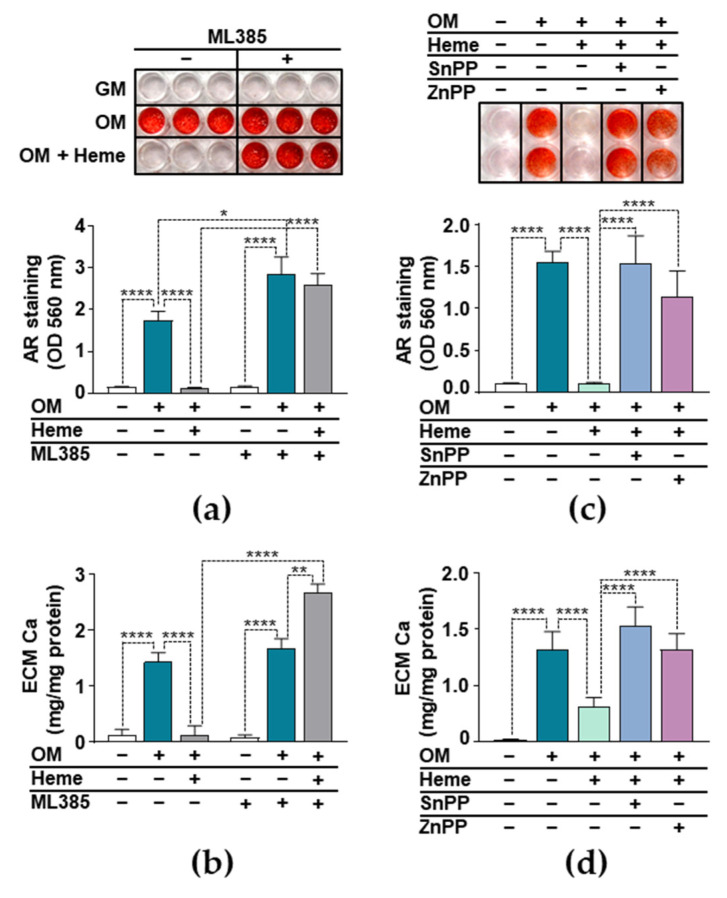
Inhibition of Nrf2 or HO-1 diminishes the anti-calcification effect of heme. (**a**,**b**) VICs (passage number 5–8) were pretreated with vehicle or ML385 (10 µmol/L) for 3 h. (**c**,**d**) VICs were pretreated with vehicle, SnPP (10 µmol/L), or ZnPP (10 µmol/L) for 3 h. (**a**–**d**) After the pretreatment, cells were maintained in GM, OM, or OM supplemented with heme (10 µmol/L). (**a**,**c**) Representative AR staining from three experiments and quantification (day 5) are shown. (**b**,**d**) The Ca content of the HCl-solubilized ECM is presented (day 5). Data are expressed as mean ± SD of three independent experiments performed in triplicate. * *p* <0.05, ** *p* <0.01, **** *p* <0.001.

**Figure 5 biomedicines-09-00427-f005:**
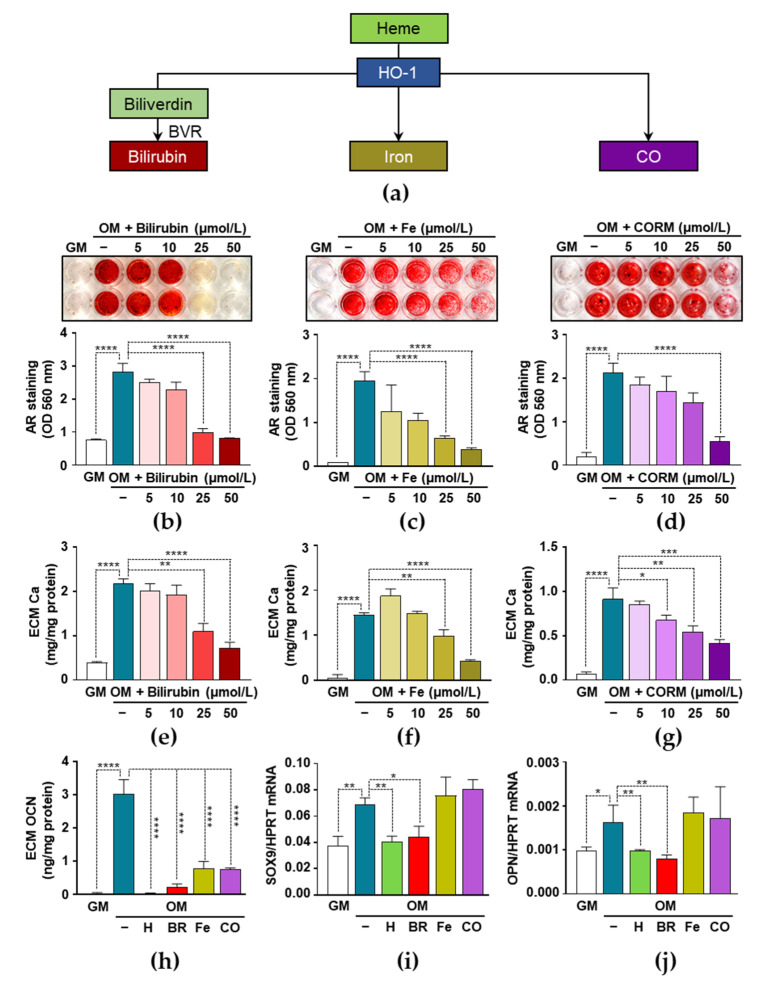
The effect of bilirubin, iron, and CO on the calcification of VICs. (**a**) Schematic of HO-1-catalyzed heme degradation. (**b**–**g**) VICs (passage number 5–8) were maintained in GM, OM, or OM supplemented with (**b**,**e**) bilirubin (BR; 5-50 µmol/L), (**c**,**f**) iron (Fe; 5-50 µmol/L), or (**d**,**g**) CO donor (CORM; 5-50 µmol/L) for 5 days. (**b**–**d**) Ca deposition in the ECM was evaluated after 5 days by AR staining. Representative images of stained plates from three independent experiments and quantification are shown. (**e**–**g**) Ca content of HCl-solubilized ECM is shown. (**h**–**j**) VICs were maintained in GM, OM, or OM supplemented with heme (H, 25 µmol/L), BR (50 µmol/L), Fe (50 µmol/L), or CORM (50 µmol/L). (**h**) OCN levels were determined in EDTA-solubilized ECM samples by ELISA (day 5). (**i**,**j**) SOX9 and OPN mRNA levels were measured by real time RT-PCR (72 h). Data are expressed as mean ± SD of three independent experiments performed in triplicate. * *p* < 0.05, ** *p* < 0.01, *** *p* < 0.005, **** *p* < 0.001.

**Figure 6 biomedicines-09-00427-f006:**
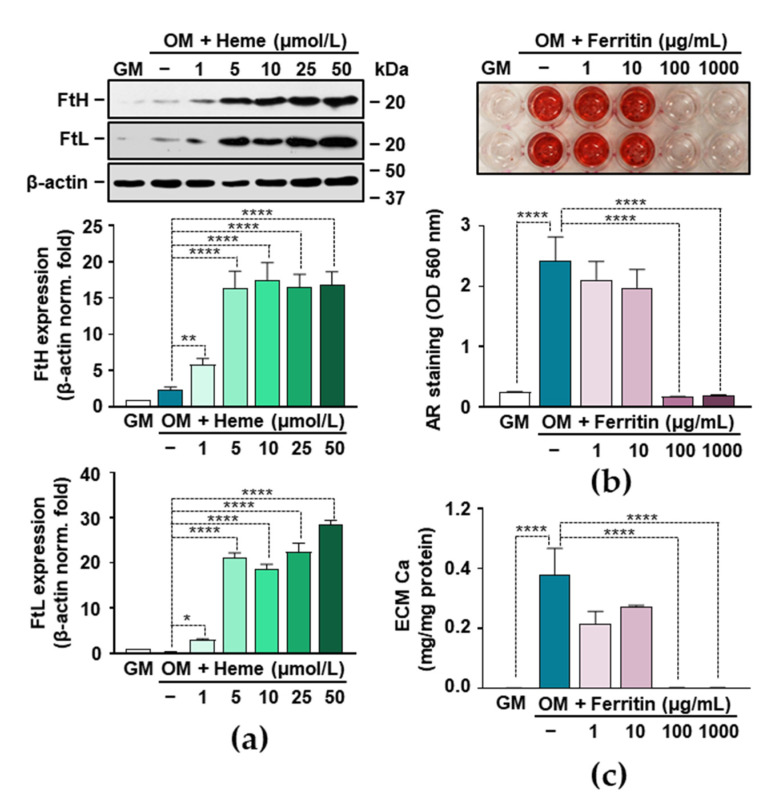
Ferritin mimics the effect of heme in inhibiting the mineralization of VICs. (**a**) VICs (passage number 5–8) were treated with GM or OM, in the absence or presence of heme (1–50 µmol/L) for 12 h. Expression of H and L chains of ferritin (FTH and FTL) were determined by Western blot. Immunoblots were reprobed with β-actin and are representative of three independent experiments. (**b**,**c**) VICs were treated with GM and OM, in the presence or absence of ferritin (1–1000 µg/mL) for 5 days. (**b**) Representative AR staining out of three independent experiments and quantification (day 5) are shown. (**c**) Ca content of HCl-solubilized ECM is shown. Results are expressed as mean ± SD from three independent experiments performed in triplicate. * *p* < 0.05, ** *p* < 0.01, **** *p* < 0.001.

## Data Availability

Original data are available upon request from the corresponding author.

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
