# Peer review of "Heme-Mediated Activation of the Nrf2/HO-1 Axis Attenuates Calcification of Valve Interstitial Cells"

_biomedicines, 2021, doi:10.3390/biomedicines9040427_

Round 1
Reviewer 1 Report
Thank you for your interesting paper
Author Response
Thank you for your effort to review our manuscript.
Reviewer 2 Report
In the present manuscript, the authors investigate the role of Heme on VICs. In particular, they identify that Heme induced Nrf2 and HO-1 expression in VICs. Heme lost its anti-calcification potential when we blocked the transcriptional activity Nrf2 or the enzyme activity of HO-1. In addition, they report that Bilirubin, CO and iron, the products of heme catabolism, as well as ferritin inhibited OM-induced 23 Ca deposition and OCN expression in VICs. They conclude that anti-calcification effect of heme is due to HO-1-catalyzed heme degradation, and ferritin.
The manuscript is interesting but several issues should be addressed before reconsideration.
- Authors report that they have used Human VICs were obtained from Innoprot (Derio, Spain). To conduct the studies at what stage/passages were the cells used?
- A clear figure reporting cell morphology of VICs should be presented
- In addition to the investigation of ALP levels, RUNX2 should also be investigated, since it is known as an important calcification marker.
- KDa for all western blots should be added.
- Is there in vivo evidence that Heme is able to reduce valve calcification? This point should be at least discussed.
- What is the translational relevance of the present findings? A separate paragraph on future perspective may be added to increase the reader's attention and curiosity.
- How Nrf2 is induced? Is it possible that keap1 is involved in this mechanism?
Minor comments
- Correct typos throughout the manuscript
Author Response
Thank you for reviewing our manuscript.
Comment 1 Authors report that they have used Human VICs were obtained from Innoprot (Derio, Spain). To conduct the studies at what stage/passages were the cells used?
Response 1 The cells were used between passage number 5 and 8. We used confluent cells in all experiments. We inserted this information into the figure legends.
Comment 2 A clear figure reporting cell morphology of VICs should be presented
Response 2 We show cell morphology in figure 1 (a) in the revised manuscript.
Comment 3 In addition to the investigation of ALP levels, RUNX2 should also be investigated, since it is known as an important calcification marker.
Response 3 Thank you for the comment. RUNX2 is the master regulator of osteogenesis and as such it plays a critical role in osteogenic differentiation of valve interstitial cells. We examined protein expression of RUNX2 in normal and calcifying conditions in VICs and found that osteogenic stimulation triggered an about 4-fold increase in the expression of RUNX2. We included this result in the revised manuscript as Figure 1f. Unfortunately we do not have data whether heme can inhibit OM-induced upregulation of RUNX2, but we have shown that heme reverses OM-induced upregulation of many other important and relevant osteogenic markers, such as SOX9 (Fig. 5i), osteopontin (Fig. 5j), ALP (Fig. 2c) and osteocalcin (Fig. 2d). We also provided solid evidence that heme inhibits OM-induced extracellular matrix calcification of VICs (Fig. 2a and b).
Comment 4 KDa for all western blots should be added.
Response 4 We added the molecular weight markers to each western blots.
Comment 5 Is there in vivo evidence that Heme is able to reduce valve calcification? This point should be at least discussed.
Response 5 We did not find any paper that would have addressed this question, therefore we completed the discussion with the following sentence:
„Further studies needed to address the effect of heme on vascular and valvular calcification in vivo.”
Comment 6 What is the translational relevance of the present findings? A separate paragraph on future perspective may be added to increase the reader's attention and curiosity.
Response 6 We completed the Conclusions and future perspective with the following paragraph: “There are pharmacological options for the upregulation of HO-1. For example a previous study showed that intravenous administration of heme arginate, a drug used in the treatment of acute porphyrias induces HO-1 in the human heart [69]. Further studies needed to investigate whether pharmacological upregulation of the Nrf2/HO-1 system, or therapeutic application of the heme degradation products could inhibit cardiovascular calcification in vivo.”
Comment 7 How Nrf2 is induced? Is it possible that keap1 is involved in this mechanism?
Response 7 In this work we induced Nrf2 with heme, which involves Keap1. We completed the introduction with the following sentences:
“Under homeostasis Nrf2 binds to its negative regulator, Kelch-like ECH-associated protein 1 (Keap1) in the cytosol [19]. This interaction initiates polyubiquitinylation and proteasomal degradation of Nrf2. Different oxidants and electrophiles induce modifi-cation of cysteine residues of Keap1 leading to the disruption of the Nrf2-Keap2 inter-action, Nrf2 stabilization and nuclear translocation. Once in the nucleus, Nrf2 binds to the antioxidant response elements located at the promoter regions a variety of antioxidant genes [19].”
„Labile heme promotes Nrf2 stabilization, via Keap1-dependent signaling mechanism and regulates the transcription of heme oxygenase-1 (HO-1), the enzyme responsible for heme catabolism [27, 28].”
Comment 8 Correct typos throughout the manuscript.
Response 8. We corrected typos.
Reviewer 3 Report
CAVS is a serious heart disease in which the differentiation of VICs into osteoblast-like cells is involved. Authors found in vitro conditions to promote the calcification of VICs. Under this condition, authors found that the supplementation of heme inhibits calcification. Because heme induces the expression of Nrf2 and HO-1, authors evaluated the effect of the inhibitors of Nrf2 and HO-1 on calcification of VICs. The calcification was recovered by the addition of these inhibitors. Further authors evaluated the effect of HO-1 products on the calcification of VICs. Supplementation of these products inhibited the calcification of VICs as well as heme. Among these products, bilirubin was the most effective inhibitor on the calcification. Because heme also induced ferritin, authors evaluated the effect of ferritin on the calcification. Externally supplemented ferritin also inhibits the calcification of VICs. These results suggest that Nrf2/HO-1 is involved in the calcification of VICs. That sounds nice and the paper is appropriate for publication in Biomedicines with major revisions as shown below.
Major comments
In figure 5, authors evaluated the effect of bilirubin on calcification of VICs. In vivo, bilirubin produced by HO-1 is further conjugated with glucuronic acid to produce conjugated (direct) bilirubin, but authors tested only the effect of unconjugated (indirect) bilirubin. Have authors checked the effect of unconjugated bilirubin?
In figure 6, authors evaluated the effect of ferritin on calcification of VICs in the range of 1-1000 μg/mL). It appears to be too high compared with the normal concentration of ferritin in blood (18-270 ng/mL). Thus this is not the result under physiological condition. In addition, what do authors consider about the contributions of iron dissociated from holo-ferritin?
Minor comments
The results of SOX9 and OPN mRNA level shown in Figure 5(i) and (j) are not explained. It must be described at line 326.
In figure 5, heme is abbreviated as "H" but it is not clear for readers.
Author Response
Thank you for reviewing our manuscript.
Comment 1 In figure 5, authors evaluated the effect of bilirubin on calcification of VICs. In vivo, bilirubin produced by HO-1 is further conjugated with glucuronic acid to produce conjugated (direct) bilirubin, but authors tested only the effect of unconjugated (indirect) bilirubin. Have authors checked the effect of unconjugated bilirubin?
Response 1 Thank you for the comment. In this work we only tested the effect of unconjugated bilirubin on calcification. The rationale behind this, is that we wanted to test the molecules that produced during HO-1 mediated heme degradation in the valve interstitial cells (VICs). In VICs heme is degraded intracellularly, producing biliverdin that is converted promptly to bilirubin by biliverdin reductase. The bilirubin forms in this reaction is unconjugated. Unconjugated bilirubin is secreted by the cells to the plasma where most of it binds to albumin. In fact albumin-bound bilirubin is the predominant bilirubin form in plasma. Bilirubin conjugation takes place in the hepatocytes in the liver. Because we worked in a cell culture medium containing 10% of FBS, we assume that most of the bilirubin was bound to albumin in our experiments that we assume, resembles mostly to the in vivo condition.
Comment 2 In figure 6, authors evaluated the effect of ferritin on calcification of VICs in the range of 1-1000 μg/mL). It appears to be too high compared with the normal concentration of ferritin in blood (18-270 ng/mL). Thus this is not the result under physiological condition. In addition, what do authors consider about the contributions of iron dissociated from holo-ferritin?
Response 2 Thank you for the valuable comment. What the reviewer refers here is the reference range for extracellular ferritin in healthy adults. It is true that the ferritin level that we used in this experiments does not represent the physiological condition. On the other hand, in iron overload patients the plasma ferritin levels can be as high as 10 μg/mL. Additionally, we have to note that the extracellular ferritins found in serum and body fluids account for a minor proportion of total ferritin.
Extracellular ferritin can be taken up by cells via specific receptors, which have been described in liver cells, on human lymphocytes and erythroblasts, adipocytes, and on various cell lines. There is no information whether ferritin is taken up by VICs, and we did not check that either. We cannot rule out the possibility that holo-ferritin was not taken up by VICs and the anti-calcification effect that we experienced is due to the iron dissociated from holo-ferritin. To clarify this issue we have inserted the following paragraph in the discussion:
“The concentration of extracellular FT in serum is below 500 ng/mL in healthy adults that is far below the FT concentration that we used in this study (1-1000 µg/mL). Extracellular FT is taken up through specific receptors found on liver cells, human lymphocytes, erythroblasts, adipocytes, and on various cell lines [64], but we lack in-formation about FT uptake by VICs. Further studies needed to investigate whether VICs take up extracellular FT and to find out whether the inhibitory effect of FT is driven or not by the iron dissociated from it.”
Comment 3 The results of SOX9 and OPN mRNA level shown in Figure 5(i) and (j) are not explained. It must be described at line 326.
Response 3 We have completed the description of the results shown in Fig. 5 with the following sentence: “Heme and BR inhibited OM-induced increase in both SOX9 and OPN mRNA expressions, whereas Fe and CO did not exhibit such protective effect (Fig. 5i and j).”
Comment 4 In figure 5, heme is abbreviated as "H" but it is not clear for readers.
Response 4 Thank you for noticing, we have corrected it.
Round 2
Reviewer 2 Report
I have no further questions.